



# Long term Observations minus Background monitoring of ground-based microwave radiometer network. Part 1: Brightness Temperatures

Francesco De Angelis[1], Domenico Cimini[2,1], Ulrich Löhnert[3], Olivier Caumont[4], Alexander Haefele[5], Bernhard Pospichal[3], Pauline Martinet[4], Francisco Navas-Guzmán[6], Henk Klein-Baltink[7], Jean-Charles Dupont[8], James Hocking[9]

[1]CETEMPS, University of L'Aquila, L'Aquila, Italy

[2]CNR-IMAA, Potenza, Italy

[3]INSTITUTE FOR GEOPHYSICS AND METEOROLOGY, University of Cologne, Cologne, Germany

[4]CNRM UMR 3589, Meteo-France/CNRS, Toulouse, France

[5]Federal Office of Meteorology and Climatology MeteoSwiss, Payerne, Switzerland

[6]Institute of Applied Physics (IAP), University of Bern, Bern, Switzerland

[7]Royal Netherlands Meteorological Institute (KNMI), The Netherlands

[8]Institut Pierre-Simon Laplace (IPSL), Université Versailles Saint Quentin, Guyancourt, France

[9]MET OFFICE, Exeter, United Kingdom

*Correspondence to:* F. De Angelis (francesco.deangelis1@graduate.univaq.it)

**Abstract.**

Ground-based microwave radiometers (MWRs) offer the capability to provide continuous, high-temporal resolution observations of the atmospheric thermodynamic state in the planetary boundary layer (PBL) with low maintenance. This makes MWR an ideal instrument to supplement radiosonde and satellite observations when initializing numerical weather prediction (NWP) models through data assimilation. State-of-the-art data assimilation systems (e.g., variational schemes) require an accurate representation of the differences between model (background) and observations, which are then weighted by their respective errors to provide the best analysis of the true atmospheric state. In this perspective, one source of information is contained in the statistics of the differences between observations and their background counterparts (O-B). Monitoring of O–B statistics is crucial to detect and remove systematic errors coming from the measurements, the observation operator, and/or the NWP model. This work illustrates a 1-year O-B analysis for MWR observations in clear sky conditions for an European-wide network of six MWRs. Observations include MWR brightness temperatures (TB) measured by the two most common types of MWR instruments. Background profiles are extracted from the French convective scale model AROME-France before being converted into TB. The observation operator used to map atmospheric profiles into TB is the fast radiative transfer model RTTOV-gb. It is shown that O-B monitoring can effectively detect instrument malfunctions. O-B statistics (bias, standard deviation and root-mean-square) for water vapor channels (22.24-30.0 GHz) are quite consistent for all the instrumental sites, decreasing from the 22.24 GHz line center (~2-2.5K) towards the high-frequency wing (~0.8-1.3K). Statistics for zenith and lower elevation observations show a similar trend, though values increase with increasing air mass. O-B statistics for temperature channels show different behaviour for relatively transparent (51-53 GHz) and opaque channels (54-58 GHz). Opaque channels show lower uncertainties (< 0.8-0.9 K) and little variation with elevation angle. Transparent



channels show larger biases (~2-3 K) with relatively low standard deviations (~1-1.5K). The Observations minus
Analysis TB statistics are similar to the O-B statistics, suggesting a possible improvement to be expected by
assimilating MWR TB into NWP models. Lastly, the O-B TB differences have been evaluated to verify the normal-
distribution hypothesis underlying variational and ensemble Kalman filter-based DA systems. Absolute values of
excess kurtosis and skewness are generally within 1 and 0.5 respectively for all instrumental sites, demonstrating
O-B normal distribution for most of the channels and elevations angles.

## 1    Introduction

The new generation of high-resolution (~1 km grid size) weather forecast models now operational over Europe
promises to improve predictions of high-impact weather, ranging from flash floods to episodes of poor air quality.
To realize this, a dense observing network is required, focusing especially on the lowest few km of the atmosphere,
so that forecast models have the most realistic state of the atmosphere for initial states and subsequent forecasts.
The United States National Research Council (NRC) recently reported that continuous planetary boundary layer
(PBL) thermodynamic observations provide a practical and cost-effective means to improve local high-impact
weather forecasting (National Research Council, 2008, 2010). However, they stated that the structure and
variability of the lower troposphere is currently not well known because vertical profiles of water vapor,
temperature, and winds are not systematically observed. This lack of observations results in the PBL being the
single most important under-sampled part of the atmosphere. While the thermodynamic state of the atmosphere is
well measured at the surface by in-situ sensors (e.g. weather stations) and in the upper troposphere by satellite
sounders, there is currently an observational gap in the PBL. Ground-based microwave radiometers (MWRs) offer
the capability to provide continuous temperature and humidity profiles in both clear- and cloudy-sky conditions
with high temporal resolution, low-to-moderate vertical resolution, and with information mostly residing in the
PBL (Cimini et al., 2006). Thus, MWR can help bridging the current observational gap in this thin layer of the
troposphere. More than thirty MWR are currently installed in Europe, most of which are operating continuously,
and the number is increasing. In this framework, MWR are candidates to supplement radiosonde and satellite
observations to feed modern numerical weather prediction (NWP) models through assimilation of their data. This
has been recently investigated in a few sporadic cases, assimilating retrieved temperature and humidity profiles
into NWP models (Cimini et al., 2012; 2014; Caumont et al., 2016). Martinet et al. (2015) illustrate the attempt to
assimilating the primary observable, i.e. brightness temperature (TB) instead of retrieved profiles, within a
simplified 1D framework, showing positive impact on the NWP forecasts in the PBL. The development of the
ground-based version of the fast radiative transfer model RTTOV, i.e. RTTOV-gb (De Angelis et al., 2016), allows
the fast simulation of ground-based MWR TB, paving the way towards the operational assimilation of MWR TB
into NWP models.
The quality of the analyses produced by data assimilation (DA) systems primarily relies on the accuracy of all
used information such as the observations, the model forecast (i.e. the background), and the observation operator
(the latter for modern DA systems such as e.g. variational systems and ensemble Kalman filters). Hence, the best
estimate of the atmospheric state is obtained only if background and observation errors are correctly described and
follow Gaussian distributions with zero mean. The representation of background and observation errors is thus
essential in the assimilation system (Waller et al., 2016).





For modern DA techniques, the observation error can be attributed to the radiometric noise, observation operator,
representativeness errors, and calibration uncertainties. The radiometric noise of the MWR is often known (~0.1-
0.2 K), well understood and approximately uncorrelated between frequency channels (Hewison, 2006a). The
occurring small correlations can be easily taken into account by observing the ambient black-body load included
in the MWR hardware. Errors arising from the observation operator uncertainty in the context of radiative transfer
modelling have been considered by De Angelis et al. (2016) for RTTOV-gb. These errors include uncertainty due
to the spectroscopy parameters (dominant and most difficult to estimate accurately) and fast model
parametrizations. The representativeness error corresponds to MWR fluctuations on smaller scales that cannot be
represented by the NWP model. In general, the contributing error terms have similar magnitudes (Hewison,
2006a). However, it is also noticeable that channels near the water vapor line at 22 GHz are dominated by
spectroscopic uncertainties, while channels most sensitive to cloud liquid water (31, 51 and 52 GHz) are dominated
by their representativeness errors, and finally the highest frequency channels (> 55 GHz), which are only sensitive
to the temperature in the lowest few hundred meters, are dominated by radiometric noise (Hewison, 2006a).
Concerning calibration errors, Maschwitz et al. (2013) quantify the uncertainty for the tipping curve calibration in
±0.1 to ±0.2 K (22-31 GHz) and ±0.6 to ±0.7 K (51-52 GHz, only be applied at high altitude sites with extremely
low water vapor content), while they show an uncertainty of ±0.9 to ±1.6 K (22-31 GHz), ±0.5 to ±1.0 K (51-53
GHz), and ±0.2 to ±0.3 K (54-58 GHz) for the liquid nitrogen calibration.
The background error covariance matrix plays an important role in data assimilation and analysis systems by
spreading the information contained in the observation both in space and between variables through cross-
correlations. A good specification of background errors is thus an essential part of any state-of-the-art data
assimilation system (Ingleby, 2001), since it affects the impact of the observations on the analysis.
The accuracy of NWP analysis systems is thus strongly dependent on appropriate statistics for both observation
and background errors. Unfortunately, those statistics are not exactly known and their determination remains a
major challenge in assimilation systems. Background errors are often determined using ensemble assimilation
systems to compute the forecast differences between each member (Brousseau et al., 2011). Differences between
observations and their background counterparts (O-B) are often used to determine observation error statistics
(Desroziers et al., 2005). The O-B monitoring in radiance space can reveal systematic errors coming from the
measurements, the radiative transfer model or the NWP forecast model (Hollingsworth et al., 1986, Stajner et al.,
2004). This approach is widely used by the satellite data assimilation community, although it may not be always
straightforward to differentiate the source of the systematic errors (Waller et al., 2016).
The bias arising from the O–B monitoring can be removed to guarantee the assumption of unbiased observations,
which is inherent to optimal estimation retrieval, such as the variational DA and ensemble Kalman filter schemes.
In this context, an accurate characterization of the MWR O-B departures represents an important step towards the
operational exploitation of the so far under-exploited MWR instruments.
This paper illustrates the analysis of MWR O–B TB differences in clear-sky conditions during one year over a
network of six instrumental sites in Central Europe. Section 2 describes the dataset and the methodology used for
this long term O-B monitoring. Section 3 discusses the results of this study, while Section 4 summarizes the
findings and draws the final conclusions.





## 2    Dataset and Methodology
### 2.1    Ground-based microwave radiometer observations
The microwave radiometer observations considered in this analysis consist of downwelling TB measured by six
commercial ground-based MWR. The MWR is a passive remote sensing instrument that measures the radiance
naturally emitted by the atmosphere at selected frequency channels in the 20–60 GHz range (Westwater et al.,
2004). MWR represent a mature technique for the retrieval of atmospheric temperature and humidity profiles as
well as integrated water vapor and liquid water path. MWRs provide retrievals in both clear- and cloudy-sky
conditions, with high temporal resolution, low-to-moderate vertical resolution, and most of information content
residing in the PBL. MWR channels near the 60 GHz oxygen complex are used to retrieve temperature profiles,
while channels near the 22.235 GHz water vapor line provide humidity and integrated water vapour information,
and are also sensitive to the column integrated liquid water content. A first attempt of MWR networking in Europe
was reported by Güldner et al., (2009) for a temporary network, while Cadeddu et al., (2013) describe the details
of the MWR network belonging to the U.S. Atmospheric Radiation Measurement program. In this study we
consider the MWR units deployed permanently at six observing sites in central Europe (JOYCE, CESAR,
LACROS, SIRTA, Payerne, and RAO – see details on Table 1). These instruments belong to different European
institutions and were chosen to be representative of the MWR technology currently deployed in Europe. In
addition, these six sites fall within the domain of the convective scale model AROME-France (Seity et al., 2011),
as shown in Figure 1. These MWR are all multichannel temperature and humidity profilers; five are manufactured
by RPG (HATPRO, Rose et al., 2005), while the remaining one (at RAO) is manufactured by Radiometrics
(MP3000A, Ware et al., 2003). HATPRO detects radiances at 14 frequency channels (22.24, 23.04, 23.84, 25.44,
26.24, 27.84, 31.40, 51.26, 52.28, 53.86, 54.94, 56.66, 57.30, and 58.00 GHz). The first seven frequency channels
are in the K-band (22 – 31 GHz), while the last 7 are in the V-band (51 – 60 GHz). MP3000A detects radiances at
12 channels (5 in the K-band and 7 in V-band). Both MWR types have elevation scanning capabilities for improved
temperature profiling in the boundary layer. The observations presented here are taken at 6 elevation angles (90.0,
42.0, 30.0, 19.2, 10.2, and 5.4°) for the HATPRO units and at 2 elevation angles (90.0 and 15.7°) for the MP3000A.
The period considered in this study extends from 1 January to 31 December 2014. During this period, the MWR
units undergo regular maintenance, including antenna radome cleaning, sanity checks, and absolute calibrations.
The maintenance strategy is currently not harmonized across the network. Absolute calibration is commonly
obtained via the cryogenic liquid nitrogen (LN2) calibration method. LN2 calibrations are typically performed
once or twice a year to correct for instrument drifts. Note that faulty calibration may happen, manifesting as
discontinuities in the time series of O-B statistics (Löhnert and Maier, 2012). Temporal matching of MWR
observations and NWP model forecasts has been obtained by selecting MWR TB records closest in time to the
model forecast time. The following O-B analysis is performed on the sample of temporal match-up observation-
model couples.
### 2.2    NWP Model
The NWP model used in this study is AROME (Seity et al., 2011). AROME has a nonhydrostatic dynamical core
inherited from the ALADIN-NH model (Bubnová et al., 1995) and physical parameterizations taken from the





research model Meso-NH (Lafore et al., 1998). In 2014, the operational configuration AROME-France covered
the domain shown in Figure 1. The model had an horizontal resolution of 2.5 km and used 60 vertical levels,
following the terrain in the lowest layers and the isobars in the upper atmosphere. The lateral boundary conditions
were provided by the global ARPEGE NWP system (Courtier et al., 1991). AROME-France used a three-
dimensional variational (3D-VAR) data assimilation system run in a rapid forward intermittent assimilation cycle,
i.e., analyses were performed every 3 hours starting from 00 UTC by assimilating all the observations available at
Météo-France in order to provide new initial states for subsequent forecasts. The background error covariance
matrices were specified through the use of an ensemble method (Brousseau et al., 2011). Data ingested by the
AROME-France DA system included observations from radiosondes, wind profilers, aircrafts, ships, buoys,
automatic weather stations, satellites, GPS stations, and both Doppler radar wind velocity and radar reflectivity
(Brousseau et al., 2014).
Temperature, humidity and pressure profiles are extracted from the AROME-France 3-hour forecasts and analyses.
AROME-France cloud liquid water profiles are not available in the dataset used for this study. The profile extracted
from the central point of the 3x3 model grid centered on each MWR site (i.e., closest in space to the MWR location)
has been used as background.
**2.3     Radiative transfer model**
MWR TB are simulated at the specific frequency channel and elevation angle from the AROME-France
thermodynamic profiles using the fast radiative transfer model RTTOV-gb (De Angelis et al., 2016). RTTOV-gb
has been developed modifying the Radiative Transfer for TOVS (RTTOV) code (version 11.2) to simulate ground-
based MWR observations, as the original RTTOV (Saunders et al., 1999) was meant to simulate downward-
viewing satellite observations only.
The Line-by-Line (LBL) model Rosenkranz (1998) has been used for the water vapour and oxygen absorption to
calculate the clear-sky transmittances needed in the RTTOV-gb regression coefficients computation (De Angelis
et al., 2016). For the RTTOV-gb training we used 83 profiles, interpolated on 101 pressure levels and carefully
chosen from a NWPSAF profile dataset to represent a wide range of physically realistic atmospheric states
(Matricardi, 2008). These 101 pressure levels (ranging from 0.005hPa to 1050hPa) have been specifically selected
for ground-based perspective to be denser close to ground (34 levels below 2km; De Angelis et al., 2016).
In this work, the RTTOV-gb simulations consider the MWR channel bandwidth through the training performed
by using LBL double sideband opacities. MWR detects radiance through narrow band-pass filters for each
frequency channel. The nominal MWR channels are characterized by the mid-frequency, which is a weighted
average over the band-pass filter. In the RTTOV-gb training we consider a rectangular filter shape characterized
by two frequencies equally weighted at the edges of the Full Width Half Maximum (FWHM). For the RPG-
HATPRO, the filter's FWHM is 0.23 GHz, except for the opaque V-band channels (0.6-2.0 GHz) (Rose et al.,
2005). The FWHM is 0.30 GHz at all the channels for the radiometrics-MP3000A (Solheim et al., 1998). RTTOV-
gb simulations take also into account atmospheric propagation effects due to Earth curvature and atmospheric
refraction (Saunders et al., 2010). In this work, RTTOV-gb does not consider the finite antenna beamwidth as this
feature is not available in the original RTTOV code. Thus, the antenna pattern, defining the region from where



radiometer antennas receive their signal, is assumed as an ideal single pencil-beam. This assumption becomes
important only at low elevation angles, e.g. up to 1-1.5 K in K-band at 5° elevation angle (Meunier et al., 2013;
Navas-Guzmán et al., 2016).
**2.4      Quality control**
Routine quality control (QC) is applied by MWR instrument operators at the individual sites, resulting in a quality
flag encoded within the datafiles. The complete data sets collected by each MWR in Table 1 have been transferred
to a common centralized server. Then, MWR observations have been quality controlled before entering the O–B
dataset.  First of all, data flagged by the sanity/rain checks provided within the instrument data stream were
discarded. In addition, we applied a cloud screening, as we intend to monitor O–B TB differences in clear-sky
only to avoid the uncertainty stemming from the forecast and absorption of cloud liquid water. Clear-sky
conditions have been selected using a two-stage screening: (i) 1-hour standard deviation of the MWR TB at 30-31
GHz ($\sigma_C$), and (ii) sky infrared temperature from the 10.5 μm infrared radiometer mounted within the MWR
housing ($T_{IR}$). Channels at 30-31 GHz are the most sensitive to clouds as they are in a gas absorption window,
where the signal is relatively insensitive to changes in atmospheric temperature and humidity. Thus, the TB
standard deviation at 30-31 GHz over a defined time period (e.g. 1-hour) can be used to indicate the presence of
liquid clouds within the MWR field of view. In addition, the infrared radiometer (not available in Payerne, SIRTA
and CESAR) is sensitive to cloud base temperature and indicates the presence of thick clouds when the infrared
temperature is high (meaning no contribution from the cold background above the cloud) (Martinet et al., 2015).
Thresholds for this screening procedure were determined in order to have a good compromise between a sufficient
data sample and a high confidence of cloudy-sky rejections (Martinet et al., 2015). Periods with $\sigma_C > 0.5$ K (Turner
at al., 2007) or $T_{IR} > -30°C$ (Martinet et al., 2015) were rejected. In addition, O–B TB differences larger than 3
standard deviations with respect to the mean difference were rejected to remove outliers (e.g. possible obstructions
or undetected cloud contamination). Table 1 reports the sample size from each instrumental site before and after
the quality control screening.

**3      Results**
An example of O-B monitoring is reported in Figure 2, showing 1-year time series of the O-B TB differences at
JOYCE for channels 22.24, 31.40, 52.28, and 58.00 GHz. Here, observations are TB measured by the HATPRO
at zenith, while background are TB simulated with RTTOV-gb from the 3-hour forecast profiles at the model grid
column closest to JOYCE. It is evident that O-B TB differences show different variance depending on the
frequency, being largest at 22.24 GHz and smallest at 58.0 GHz. O-B TB differences show to be quite steady, with
the exception of channel 31.40 GHz; here a large difference (up to 10 K) is evident until 3 June 2014 (Julian day
154). This misbehavior was later confirmed by the instrument operator and it was attributed to a faulty calibration.
In fact, 3rd June 2014 corresponds to the date of the new LN2 absolute calibration at JOYCE, after which the
observation comes closer to background again. This demonstrates that the O-B monitoring is able to detect
instrument malfunctions, and it should be implemented and performed at each MWR site as part of its quality
control procedure. Similar misbehaviors were detected and later confirmed by instrument operators at other sites.





Specifically: (i) at CESAR at all the channels below 54 GHz between 15 June and 18 September, 2014,
corresponding again to a period after a faulty calibration; (ii) at CESAR at channel 22.24 GHz and elevation angles
below 42 degrees, due to radio frequency interference (RFI) leaking into the channel bandpass filter; (iii) at
Payerne at 26.24 GHz for the whole period, due to an unknown malfunction possibly related to hardware
components causing large observed TB variations.
Figure 3 shows the O-B TB statistics for the 6 instrumental sites at zenith (i.e. 90° elevation angle). The reported
bias, standard deviation (Std) and root-mean-square (RMS) are computed from the QC data set. Note that periods
of instrument malfunctions have been removed in JOYCE (before Julian day 154) and CESAR (Julian days
between 165 and 261) by discarding the data before computing the statistics. The 26.24 GHz channel misbehavior
in Payerne has not been removed because the source is still unidentified and it also affects the whole dataset.
Accordingly, bias, Std and RMS for this channel show a peak, reaching -3, 2, and 4 K, respectively. Std statistics
for the K-band channels show very similar behaviour from site to site, decreasing from the line center towards the
high-frequency wing. This may suggest that the O-B difference is mostly due to an uncertainty in the humidity
profile forecast. For example, at JOYCE the Std ranges from 1.6 K at 22.24 GHz to 0.7-0.8 K at 27.84/31.40 GHz.
Note that channels close to the 22.24 GHz line center show the highest values of TB and the highest dynamic range
in clear-sky conditions. Thus, the large absolute uncertainty at these channels may correspond to similar relative
accuracy when compared to the other K-band channels.
The maximum RMS and biases are located at 22.24 GHz at all the sites except RAO where they are at 23.04 GHz
(around 2.5 and 2.0 K, respectively). At JOYCE the bias and RMS range respectively from 0.9 and 1.8 K at 22.24
GHz to 0.1 and 0.7 K at 27.84 GHz. Albeit one may not expect the variability to increase in the window channel,
we see slightly larger differences at 31.40 GHz with respect to 27.84 GHz (bias and RMS equal to 0.8 and 1.3 K,
respectively), which may be attributed to few undetected cases of cloud contamination. Similar statistics are
reported at all sites except LACROS. At LACROS we see similar standard deviations, but larger bias and,
consequently, RMS (ranging from 1.7 to 3.6 K in K-band, with maximum value at 22.24 GHz). The reason for
these larger biases is still under investigation.
O-B statistics at V-band show different behavior at lower frequency (i.e. transparent) and higher frequency (i.e.
opaque) channels. Opaque channels (54-58 GHz) show low bias, std, and RMS (all within 0.9 K) as the atmosphere
is opaque due to oxygen and therefore water vapour and the effect of clouds on observed TB is negligible at these
channels. Transparent channels (51-53 GHz) show rather large biases (2-3 K and up to 5 K in Payerne) with
relatively low std (1.0-1.5 K). Bias values of the same order of magnitude for the 51-53 GHz range were previously
reported (Hewison et al. 2006b; Löhnert and Maier, 2012, Martinet et al. 2015, Blumberg et al., 2015; Navas-
Guzmán et al. 2016), employing MWR of different types and manufacturers. Large biases at lower V-band
channels (50-54 GHz) are likely due to a combination of systematic uncertainties stemming from inaccurate
instrument bandpass characterization, calibration and absorption model. In fact, these channels are located on a
steep shoulder of the $O_2$ absorption complex, and thus are sensitive to uncertainty in band-pass modeling. In
addition, they suffer from larger calibration uncertainty, due to the relative low opacity as well as larger radiative
transfer model errors due to the lack of well calibrated data usable for tuning spectroscopic parameters. However,
it is important to note that the standard deviation remains below 1 K allowing for an easy bias correction. In this
study, a bias correction based on simulated TB computed from clear-sky NWP model profiles can thus be applied
on the measurements.



Figure 4 shows O-B statistics for zenith MWR observations in JOYCE, before and after such a bias correction.
The correction values were computed by a previous work considering the DWD COSMO-DE model (Baldauf et
al., 2011), using forecasts not older than 3h at the closest vertical column to JOYCE. These columns provide
temperature, pressure and humidity profiles necessary for forward calculations of TB. All clear sky observations
between two absolute calibrations have been used to compute the V-band biases by considering simultaneous
observations and forward modeled TB. This approach assumes a constant bias between two adjacent calibrations,
which has been justified for a HATPRO system by Löhnert and Maier (2012). In this way, the TB biases in the V-
band are decreased from 1-1.5 K to 0.1-0.5 K. between 51 and 53 GHz (Figure 4). Note that the bias correction is
applied to the V-band channels only, since the humidity uncertainty affecting the model and the colocation is
deemed too high for providing a reliable bias correction for the K-band channels. A different approach using NWP
model output to adjust microwave observations for operational applications is discussed by Güldner (2013).
Ideally, the bias correction should be computed using the same NWP and radiative transfer models used for the O-
B, as done in operational systems. Conversely, the bias correction applied in Figure 4 is derived from COSMO
while the O-B is performed with AROME. Also the radiative transfer models are different, though both adopting
the atmospheric absorption model of Rosenkranz (1998). However, even though the adopted bias correction
procedure is not perfect, we already see a significant improvement in the O-B statistics demonstrating that a bias
correction would remove most of the systematic errors at 51-53 GHz, on the assumption of consistent NWP and
radiative transfer models.
Table 2 reports the O-B TB mean differences (i.e., biases) and their 95% confidence intervals, for each instrumental
site at zenith.
Observations at different elevation angles allow to check the robustness of the previous results. Figure 5 shows
the statistics at JOYCE at elevation angles 90.0, 42.0, 30.0, 19.2, 10.2, and 5.4°.  Results at K-band show similar
tendencies at lower elevation angles (panels B to F). However, the variability increases with decreasing elevation
angle, because uncertainty in the AROME-France humidity profile gets amplified with increasing air mass. This
happens also due to a stronger TB signal resulting in larger absolute noise. Statistics follow a similar trend up to
19.2° elevation angle, where RMS reaches 6 K. Larger differences are found at 10 and 5° elevation angles in K-
band (biases, Std and RMS respectively up to 16, 8 and 18 K at 5°) probably due to: (i) the current version of
RTTOV-gb is not designed for elevation angles lower than 15° (De Angelis et al., 2016), as simulations at low
elevation angles were not necessary in the original satellite perspective; and (ii) the violation of the homogeneity
assumption which needs to be satisfied when using low elevation angles. This may also be due to the fact that 10°
and 5° are outside the elevation angle range used in the RTTOV-gb training configuration (elevation angle set
between 90° and 16°) (De Angelis et al., 2016). Moreover, while RTTOV-gb considers earth curvature, band width
and atmospheric refraction (as explained in Section 2.3), it currently does not take into account the antenna beam
width; this aspect can cause large biases between simulations and observations at very low elevation angles.
Statistics at V-band opaque channels show little variation with elevation angle. The zenith systematic O-B
differences in the 52.28 and 53.86 GHz channels decrease with decreasing elevation angle due to the fact that
atmosphere becomes more and more opaque. However, the systematic difference at 51.26 GHz stays between 1
and 2 K, independent of elevation angle. Here, the systematic offset at zenith (possibly due to calibration
uncertainty)  is probably taken over by effects of not considering the antenna beam width at low elevation angles





(see Meunier et al. 2013, Figure 14). The statistics of random uncertainty (i.e. Std) follow a similar trend with
elevation angle at all the instrumental sites (Figures for all sites are reported in the supporting document).
Figure 6 shows the statistics of O-B differences as well as Observation minus Analysis (O-A), in which the Arome
analysis is used instead of the 3-hour forecast as background. O-B and O-A statistics at JOYCE, elevation angles
90° and 19.2°, are shown. O-B and O-A biases are similar at both elevation angles in the K-band and in the V-
band more transparent channels. Thus, forecast and analysis compare almost equally to observations; this indicates
that the newly assimilated data did not bring significant information to the analysis with respect to forecast in
terms of MWR observables. Assuming the observations as the reference, and considering that MWR uncertainty
for transparent channels is typically smaller than the RMS in Figure 6, this may suggest that there is useful
information in MWR data for improving NWP data assimilation. Considering V-band opaque channels, we note
smaller biases for O-A than O-B differences (up to 40% smaller at 19.2°). These channels are mostly sensitive to
temperature profile in the PBL, and thus this result suggests that most of the 3h-forecast errors point toward the
PBL. PBL is indeed the atmospheric layer where most of the information provided by MWRs is located, though
this may be redundant with that of other assimilated observations (e.g., radiosondes). Quantification of the
information brought by MWR into NWP data assimilation will be the subject of future research.
Note that O-A standard deviations are slightly lower than or equal to the corresponding O-B Std at all the frequency
channels; this is consistent with the assumptions that the analysis variance is lower than or equal to the background
variance, and the observation errors and the model errors (either analysis or background) are independent. These
assumptions are usually made in modern DA techniques.
In addition to unbiased observations, another hypothesis common to variational and ensemble Kalman filter-based
DA systems is that the observations and background errors are Gaussian, which implies that the distribution of the
O-B TB differences is Gaussian. This assumption has been verified by exploiting excess kurtosis and skewness
scores. Kurtosis can be formally defined as the standardized fourth population moment about the mean of a specific
distribution. The normal distribution has a kurtosis of 3, thus the "excess kurtosis" is usually used (i.e., kurtosis −
3). A distribution with positive excess kurtosis has heavier tails and a higher peak than the normal distribution,
whereas a distribution with negative excess kurtosis has lighter tails and is flatter. Skewness is formally the third
central moment of the specific distribution, divided by the cube of its standard deviation, and it is a measure of
symmetry, or more precisely, the lack thereof. A positive skewness value indicates positive (right) skew; a negative
value indicates negative (left) skew. The higher the absolute value, the greater the skew.
For both excess kurtosis and skewness, a normal distribution should return a score of 0. In general, fair
approximations to normal distribution should have skewness and excess kurtosis within -1 and +1 (Bulmer, 1979).
Figure 7 shows two histograms of O-B TB differences at 90° elevation angle, for JOYCE (58.00 GHz) and Payerne
(52.28 GHz). The histogram at JOYCE shows a distribution approximately Gaussian, with excess kurtosis 0.15
and skewness -0.07. Conversely, the distribution at Payerne has heavier tails (excess kurtosis 1.86) than Gaussian
and moderate asymmetry (skewness 0.72). Figures 8 and 9 show respectively excess kurtosis and skewness as
function of frequency, at each instrumental site and for elevation angles 90.0, 42.0, 19.2 (15.7 for RAO) and 10.2°.
In general, excess kurtosis is within ±1, demonstrating fair approximation to Gaussian error. Excess kurtosis above
2 is reported for Payerne at 51-53 GHz and 90°-42° elevation and for CESAR at 22.24 GHz and 42° elevation.
These same channels are also characterised by large O-B TB statistics, as shown in Figure 3 and in the supporting
document. Above 10° elevation, absolute values of excess kurtosis slightly exceeding 1 are reported for RAO at





23.04 GHz and 15.7°, and for LACROS at 51.26 GHz and 19.2°. At 10°, kurtosis around 1.3-1.5 is reported for
Payerne (31.40, 51.26 and 52.28 GHz) and for CESAR (51.26 GHz). In general, the absolute value of the skewness
is within 0.5 in K-band (22-28 GHz) and in the V-band opaque channels (54-58 GHz) at each instrumental site,
meaning approximately symmetric distributions. The only exception is CESAR at 22 GHz and 42°, where
skewness is 1.2. Larger skewness are reported for more transparent channels (31, 51 and 52 GHz), in particular
for Payerne (52 GHz, elevation angles 90 and 42°, up to 0.9), and for Joyce (51 GHz, 10° elevation angle, up to
1.1), demonstrating moderate asymmetry. In summary, of the 328 channel and elevation angle combinations that
were evaluated, only 4.2% (0.6%) showed an absolute value of excess kurtosis (skewness) larger than 1. Among
these are the channels that showed large O-B statistics and are thus suspect of instrumental misbehaviour.

## 11   4    Summary and conclusions

This work illustrates the first Observations minus Background (O–B) analysis of ground-based brightness
temperature (TB) observations from an European network of six microwave radiometers (MWR) over a 1-year
period (2014). Statistics of the differences between MWR observations and their NWP model background
counterparts can be used to shed light on observation and background errors. The knowledge of these errors is
crucial for data assimilation because observations and short-term model forecasts are the primary sources of
information used to produce analyses. Moreover, the O-B monitoring is essential to detect and possibly remove
any systematic errors coming from the MWR measurements, the radiative transfer model or the NWP model
forecast.
In this analysis, observations are MWR TB measured by two types of commercial MWR (RPG-HATPRO and
Radiometrics-MP3000A). Background counterparts are TB simulated with the fast radiative transfer model
RTTOV-gb from the AROME-France 3-hour forecasts and analyses. Quality control and clear-sky selection are
performed with a three-stage screening based on the 1-hour standard deviation of the MWR TB at 31 GHz, the
infrared radiometer TB, and the quality/rain flag provided by the manufacturer.
It is shown how O-B monitoring can be used to detect instrument malfunctions by exploiting the timeseries of the
O-B TB differences. The results strongly suggest an operational implementation of this monitoring at all sites
deploying a MWR as part of the quality control procedure.
Observations minus background statistics are quite consistent between the instrumental sites. They decrease in K-
band at zenith from the 22.24 GHz line center (Std~1.5-2.0 K) towards the high-frequency wing (Std~0.5-1.0 K).
V-band opaque channels (54-58 GHz) show low statistics (RMS within 0.8-0.9 K) due to the saturation and the
dependence to only temperature. V-band more transparent channels (51-53 GHz) show large biases (up to 5 K in
Payerne) with relatively low std (1.0-1.5 K), demonstrating that these biases can be effectively removed by
applying a bias correction based on TB simulated from a NWP model (if the forecast errors are within the expected
accuracy).
Statistics at K-band increase with decreasing elevation angle, following a similar trend. Large differences are found
at low (5-10°) elevation angles (RMS up to 20 K) due to atmospheric inhomogeneity and known RTTOV-gb
limitations at elevation angles below 15°. Statistics at V-band opaque channels show only small variations with
the elevation angle. The O-B mean differences decrease at 52.28 and 53.86 GHz and increase at 51.26 GHz with
decreasing elevation angles.





The Observations minus analysis TB statistics are similar to the O-B, except for a bias deflection (up to 40%) in
the V-band opaque channels, especially at low elevation angles. This suggests the possible level of improvement
that may be expected by assimilating MWR TB into NWP models, at least for boundary layer temperature
profiling.
The Gaussian error assumption, typical of variational and ensemble Kalman filter-based DA systems, has been
evaluated by computing excess kurtosis and skewness scores of the O-B TB distributions. Among the evaluated
angle/channel combinations, excess kurtosis and skewness are typically within respectively 1 (95.8%) and 0.5
(99.4%). This demonstrates that O-B TB distributions are typically Gaussian with good approximation. Larger
scores (excess kurtosis and skewness above 2 and 1) are reported for Payerne at 51-53 GHz and high elevation
angles (90-42°), and for Cesar at 22.24 GHz and 42° elevation angle. These scores result in O-B TB distributions
with moderate asymmetry and heavier tails than Gaussian, possibly due to instrument malfunction or radio
frequency interference.
In conclusion, the presented O-B analysis demonstrated the typical operational performances of a prototype
network of six MWR in Europe, showing:
1) Robust and mature technology, suitable for operational use;
2) Continuous TB observations, typically stable and reliable, whose quality can be monitored remotely;
3) Consistent O-B statistics throughout the network;
4) Moderate O-B systematic differences that can be effectively addressed through bias correction;
5) Typically Gaussian O-B distributions;
This work provides a comprehensive characterization of the MWR O-B statistics and distributions that may serve
as a reference for the other MWR currently deployed in Europe and world-wide, including commercial (e.g. Attex
MTP-5) and research types (e.g. TEMPERA, Navas-Guzman et al., 2016). It also represents a step towards the
operational exploitation of ground-based MWR, so far under-exploited instruments that may play a crucial role in
the accurate characterization of boundary layer thermodynamics into NWP models.

**5 Acknowledgements**
This work has been stimulated through the COST Action ES1303 (TOPROF), supported by COST (European
Cooperation in Science and Technology). Part of the work was supported by the EU H2020 project GAIA-CLIM
(Ares(2014)3708963/Project 640276). The JOYCE observations and data analysis have been supported through
the German Science Foundation DFG under grant LO 901/7-1. RAO, LACROS, JOYCE and CESAR observations
have also been supported through the German BMBF research initiative HD(CP)[2] under grants FKZ01LK1209A-
E as well as FKZ01LK1502A. Thanks to the Leibniz-Institute for Tropospheric Research (TROPOS) in Leipzig
and the Deutscher Wetterdienst's (DWD) Richard-Aßmann-Observatorium (RAO) in Lindenberg for providing
the data respectively for LACROS and RAO stations. The authors would like to thank the technical and computer
staffs of SIRTA Observatory for taking the observations and making the data set easily accessible. Comments on
early drafts by Christine Knist (DWD) were greatly appreciated. The authors would like to acknowledge all the





members of the TOPROF Working Group 3 for providing fruitful discussion on the instrument performances and
O-B analysis.

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


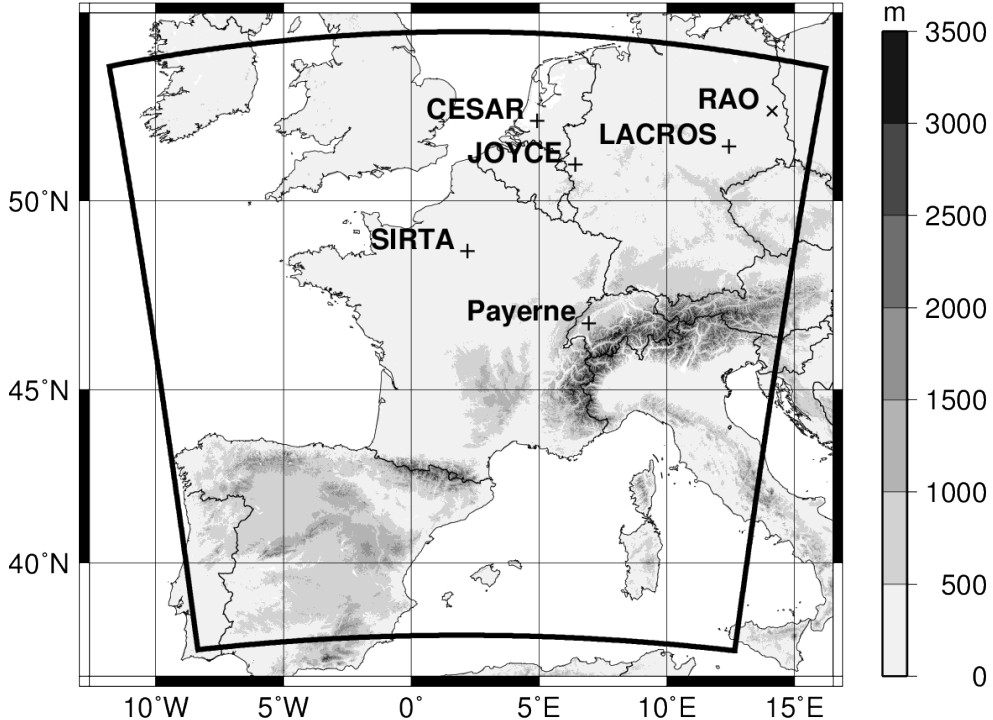

2
3  **Figure 1: Topography (m) and domain of AROME-France (large area delineated by solid black line). Locations of**
4  **MWR sites are also shown (+ and × indicate respectively HATPRO and MP3000A).**



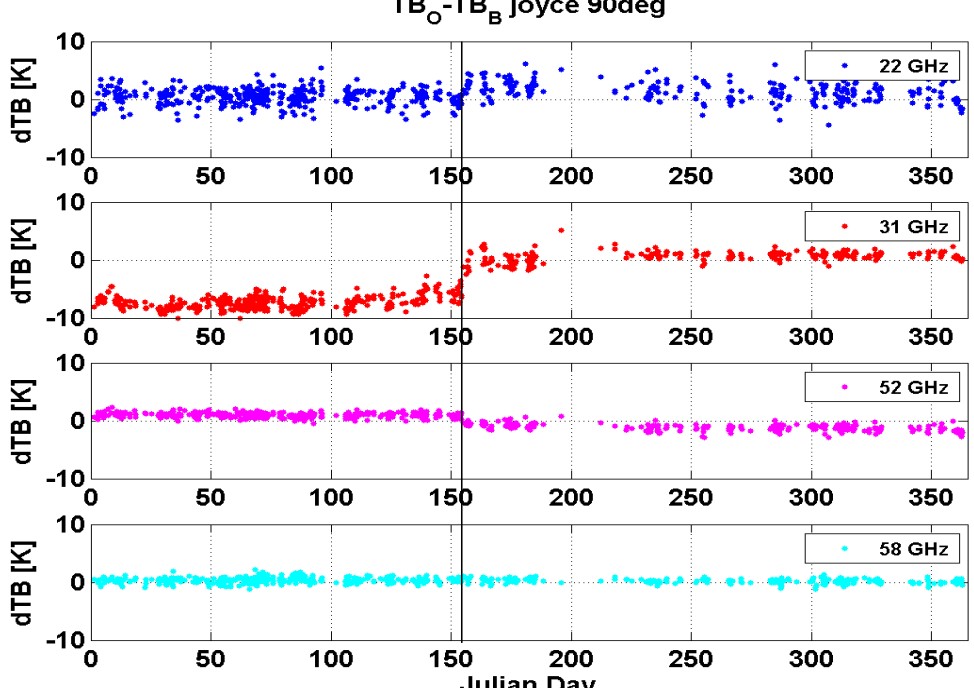

**Figure 2: Timeseries of the O-B TB differences at JOYCE; from top to bottom: channels 22.24 (blue dots), 31.40 (red**
**dots), 52.28 (magenta dots), and 58.00 GHz (cyan dots). The black solid line represents the date of the new calibration**
**(June 3rd, 2014).**





**Figure 3: Statistics of the differences between observations and background TB. Observations are TB measured by ground-based MWR. Background counterparts are TB simulated with RTTOV-gb from AROME-France 3-hour forecast profiles in clear-sky conditions at zenith. Panels A, B, C, D, E, and F refer respectively to JOYCE, LACROS, Payerne, SIRTA, CESAR and RAO. Biases are shown with black lines, standard deviations with red lines and RMS with blue lines.**





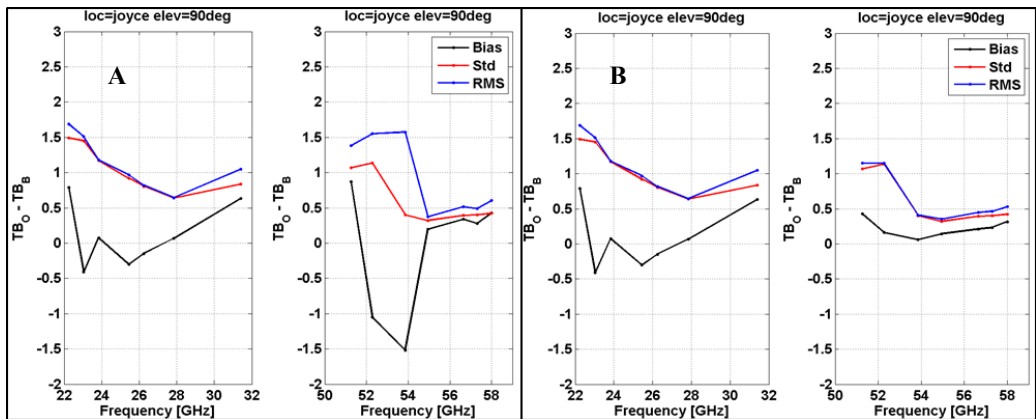

3    **Figure 4: Statistics of the differences between observations and background TB, as in Figure 3. Here, results from**

4    **JOYCE at zenith are shown. Panels A, and B, refer respectively to before and after the bias correction with the COSMO-**

5    **DE model. Biases are shown with black lines, standard deviations with red lines and RMS with blue lines.**





**Figure 5: Statistics of the differences between observations and background TB, as in Figure 3. Here, results from JOYCE at different observing angle are shown. Panels A, B, C, D, E, and F refer respectively to 90, 42, 30, 19.2, 10.2, and 5.4° elevation angle. Biases are shown with black lines, standard deviations with red lines and RMS with blue lines.**



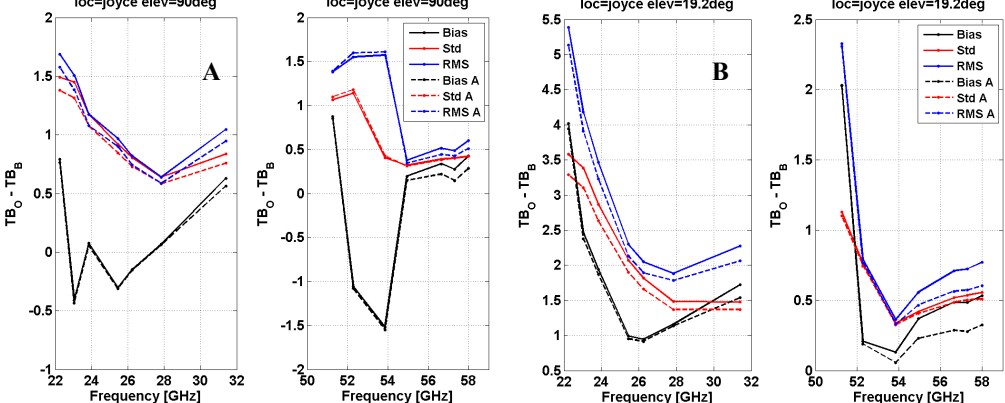

**Figure 6: Statistics of the differences between TB observations and model background (solid lines), and TB observations and model analysis (dashed lines). Simulated TB are computed with RTTOV-gb respectively from AROME-France 3-hour forecast (solid lines) and AROME-France analyses (dashed lines) profiles in clear-sky conditions for Joyce at zenith. Panels A, and B, refer respectively to 90 and 19.2° elevation angle. Biases are shown with black lines, standard deviations with red lines and RMS with blue lines.**

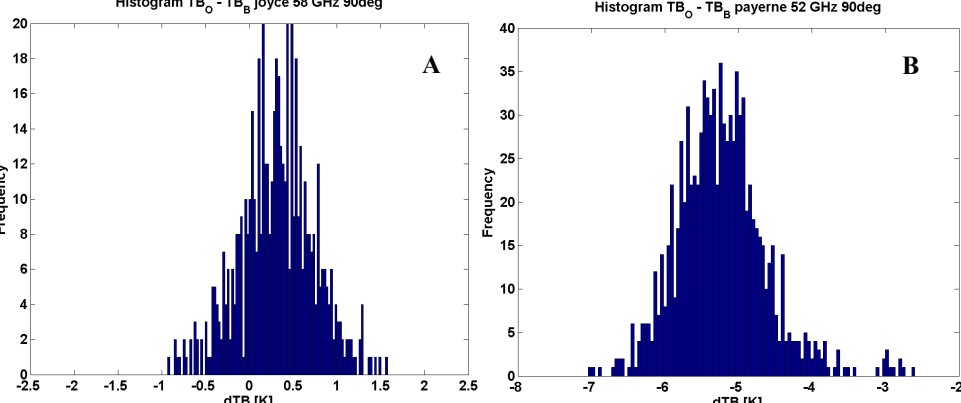

**Figure 7: histograms of the O-B TB differences. Panel A refers to JOYCE at 58.00 GHz and 90° elevation angle, while Panel B refers to Payerne at 52.28 GHz and 90° elevation angle.**





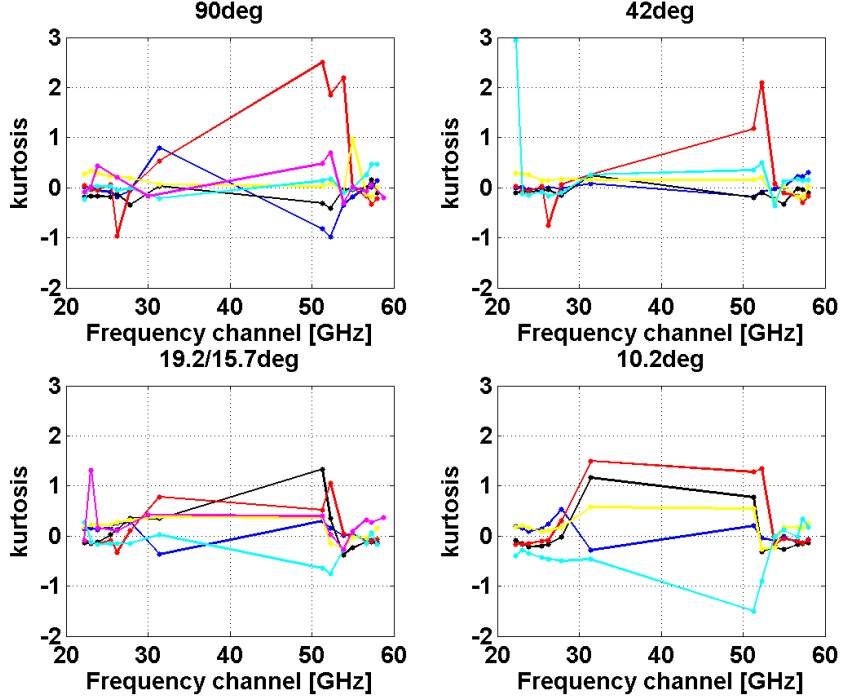

2  **Figure 8: Excess kurtosis as function of frequency, for elevation angles 90.0, 42.0, 19.2 (15.7 for RAO) and 10.2°. Scores**

3  **for JOYCE, LACROS, Payerne, SIRTA, CESAR and RAO are reported respectively in blue, black, red, yellow, cyan,**

4  **and magenta.**



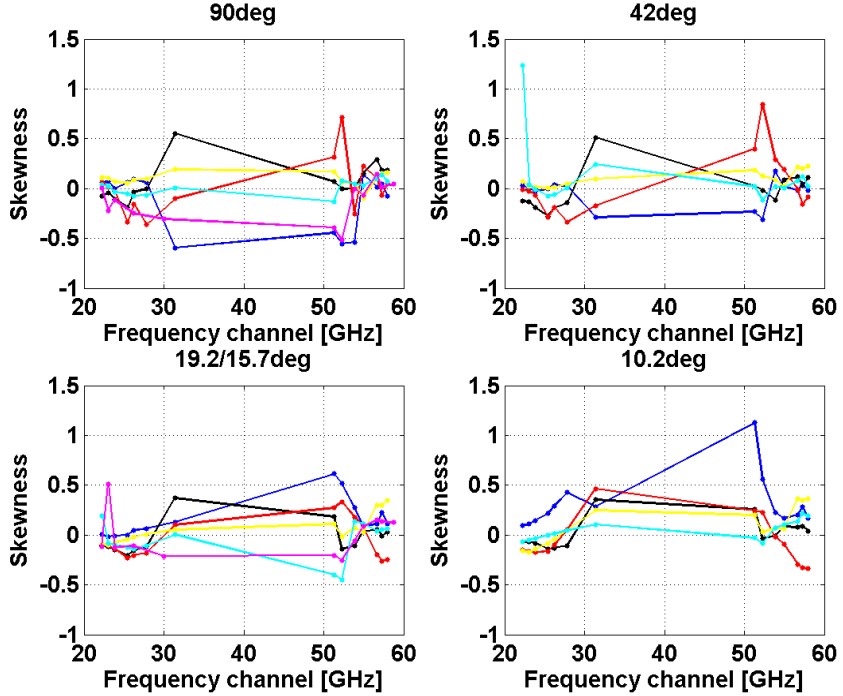

**Figure 9: Skewness as function of frequency, for elevation angles 90.0, 42.0, 19.2 (15.7 for RAO) and 10.2°. Scores for**
**JOYCE, LACROS, Payerne, SIRTA, CESAR and RAO are reported respectively in blue, black, red, yellow, cyan, and**
**magenta.**
**Table 1: Sample size at all the instrumental sites before and after the quality control screening. Position and height of**
**each instrument are reported. For HATPRO only, the generation family is also reported (G5 is currently**
**commercialized).**

| LOCATION | LAT | LON | HEIGHT(m) | MWR | PRE-SCREENING | POST-SCREENING |
|---|---|---|---|---|---|---|
| JOYCE | 50.91 | 6.41 | 111 | HATPRO G2 | 602 | 557 |
| LACROS | 51.35 | 12.43 | 125 | HATPRO G2 | 542 | 502 |
| Payerne | 46.82 | 6.95 | 491 | HATPRO G1 | 1087 | 955 |
| SIRTA | 48.80 | 2.36 | 156 | HATPRO G2 | 1022 | 923 |
| CESAR | 51.97 | 4.93 | -0.7 | HATPRO G1 | 988 | 664 |
| RAO | 52.21 | 14.12 | 125 | MP3000A | 709 | 680 |



1    **Table 2: biases of O-B TB differences and their 95% confidence intervals, for all the instrumental sites at zenith. Values**

2    **at RAO (MP3000A) are reported on the colomn of the closest HATPRO frequency channel.**

| Chan (GHz) | 22.24 | 23.04 | 23.84 | 25.44 | 26.24 | 27.84 | 31.40 | 51.26 | 52.28 | 53.86 | 54.94 | 56.66 | 57.30 | 58.00 |
|---|---|---|---|---|---|---|---|---|---|---|---|---|---|---|
| JOYCE | 0.791 | -0.411 | 0.075 | -0.303 | -0.148 | 0.068 | 0.630 | 0.874 | -1.052 | -1.519 | 0.196 | 0.337 | 0.275 | 0.428 |
| | ±0.124 | ±0.121 | ±0.098 | ±0.077 | ±0.067 | ±0.053 | ±0.114 | ±0.089 | ±0.095 | ±0.034 | ±0.027 | ±0.033 | ±0.033 | ±0.035 |
| LACROS | 3.288 | 2.776 | 2.179 | 1.698 | 1.688 | 1.605 | 2.425 | -0.341 | -2.536 | 0.026 | -0.179 | -0.009 | 0.066 | -0.004 |
| | ±0.130 | ±0.121 | ±0.105 | ±0.075 | ±0.063 | ±0.049 | ±0.093 | ±0.047 | ±0.039 | ±0.030 | ±0.034 | ±0.042 | ±0.044 | ±0.044 |
| Payerne | 1.666 | 1.105 | 0.995 | -0.085 | -3.156 | 0.029 | 0.409 | 3.941 | -5.230 | 1.978 | -0.781 | -0.017 | 0.082 | -0.090 |
| | ±0.107 | ±0.103 | ±0.087 | ±0.061 | ±0.162 | ±0.050 | ±0.038 | ±0.051 | ±0.039 | ±0.019 | ±0.020 | ±0.029 | ±0.035 | ±0.031 |
| SIRTA | 2.243 | 1.645 | 1.810 | 1.161 | 1.105 | 1.203 | 0.943 | -1.284 | -3.425 | -2.301 | -0.153 | -0.064 | 0.012 | 0.121 |
| | ±0.110 | ±0.106 | ±0.089 | ±0.061 | ±0.053 | ±0.043 | ±0.039 | ±0.051 | ±0.046 | ±0.023 | ±0.033 | ±0.035 | ±0.037 | ±0.037 |
| CESAR | 1.615 | 0.840 | 0.293 | -0.190 | -0.061 | 0.065 | 0.243 | 0.838 | -0.883 | 1.139 | -0.632 | -0.279 | -0.300 | -0.233 |
| | ±0.150 | ±0.124 | ±0.117 | ±0.073 | ±0.064 | ±0.053 | ±0.050 | ±0.062 | ±0.043 | ±0.029 | ±0.029 | ±0.033 | ±0.032 | ±0.034 |
| RAO | 1.588 | 2.032 | 0.647 | | 0.821 | | 0.940 | -0.549 | -2.706 | -2.429 | -1.262 | -0.905 | -0.932 | -0.831 |
| | ±0.122 | ±0.117 | ±0.086 | | ±0.063 | | ±0.060 | ±0.057 | ±0.048 | ±0.031 | ±0.034 | ±0.039 | ±0.039 | ±0.045 |

