# Peer review of "Long term Observations minus Background monitoring of ground-based microwave radiometer network. Part 1: Brightness Temperatures"

_Atmospheric Measurement Techniques, 2017_

## Author Comment (AC1) · 1 May 2017

Thanks very much for your positive feedback and constructive suggestions. We really appreciated your input. Let us provide more information on each topic you have raised.

1) 3-hour forecast is used and it is not sure if the model has well spin up after 3 hours. In the future, also for a more global study, ECNWF analysis and forecast would be preferable.

The spin-up time of AROME has been studied thoroughly. For AROME, the spin-up time has been estimated to less than 2 hours, by looking at the time evolution of the root mean square of surface pressure tendency (Brousseau et al. 2008).

[Figure]

Future work will extend the present prototype study to provide a global perspective, exploiting the output of global models. In this perspective, ECMWF as well as NCEP data have been used as background in a previous O-B study in the retrieval space (Cimini et al., 2010).

2) suggest to include radiometers in other climates and geological locations, e.g. those in American, southern China and Korea/Japan

As suggested, future work will be dedicated to a global study, including additional sites and MWR types selected from the MWRnet members: http://cetemps.aquila.infn.it/mwrnet/

3) a longer period of data would be useful for studying the long term behaviour of the radiometers

Extending the O-B study to a longer period is surely useful for studying the long term behaviour of MWR systems. However, following the manufacturers' instructions, MWR systems are calibrated regularly (typically every ïA¿6 months, depending on sites). Instrumental drifts should be mitigated by the calibration procedure. In this study we aimed at providing typical O-B statistics that one may expect from MWR and NWP models. Therefore, we deemed one year as a sufficiently long period for the above aim.

References:

Brousseau, P., F. Bouttier, G. Hello, Y. Seity, C. Fischer, L. Berre, T. Montmerle, L. Auger, S. Malardel, 2008: A prototype convective-scale data assimilation system for operation: the Arome-RUC. HIRLAM Tech. Rep., 68, 23-30, URL: http://tinyurl.com/Brousseau-et-al-2008-pdf

Cimini D., E. R. Westwater, and A. J. Gasiewski, Temperature and humidity profiling in the Arctic using millimeter-wave radiometry and 1DVAR, IEEE Transactions on Geoscience and Remote Sensing, Vol. 48, 3, 1381-1388, 10.1109/TGRS.2009.2030500,

2010.

---

## Referee Comment (RC1) · M. Cadeddu (Referee) · 25 May 2017

The comment was uploaded in the form of a supplement: http://www.atmos-meas-tech-discuss.net/amt-2017-112/amt-2017-112-RC1-supplement.pdf
* * *

---

## Author Response (AR1)

**REPLY TO Dr. Maria Cadeddu**

The authors are grateful to the referee for the positive feedback and the constructive comments. We revised the manuscript accordingly. Our point by point replies are shown in red hereafter, while modifications to the text are highlighted in yellow within the revised manuscript.

**1) Page 3 lines 11-13: "…while channels most sensitive to cloud liquid water (31, 51, and 52) are dominated by their representativeness errors". I am not sure I agree with this statement. Channels at 31, 51 and 52 GHz are very sensitive to the water vapor continuum as well and to the spectroscopy of the 50-60 GHz line complex (for example the mixing coefficients).**

Agreed. The following sentence has been included in the revised manuscript:

"Moreover, channels at 31, 51 and 52 GHz are also sensitive to the water vapor continuum as well and to the spectroscopy of the 50-60 GHz line complex"

**2) Page 3 line 16: "0.5 – 1 K (51-53 GHz)". This uncertainty for LN2 calibration is very optimistic in most field conditions.**

L2N calibration uncertainties reported in the manuscript refer to the results of Maschwitz et al. (2013). In this study, a theoretical error propagation based on the HATPRO specifications was developed. Albeit to our knowledge Maschwitz et al. (2013) presented the most complete uncertainty analysis of LN2 calibration available in the literature, we acknowledge it may not cover all error sources affecting field conditions, such as condensate on the radome, spurious reflections, and receiver sensitiviy drifts. Thus, we concur with the referee this may slightly underestimate the total uncertainty. We have added a warning in the revised text to highlight this.

**3) Page 5 Section 2.3: It may be a good idea to give a quick summary of how the fast radiative transfer model RTTOV-gb works and why one needs to train it.**

Agreed. The following text on how the fast radiative transfer model RTTOV-gb works has been introduced in the revised manuscript:

"Fast radiative transfer models perform simplified calculations of the atmospheric radiances by parameterizing the atmospheric transmittances. Accurate transmittances, computed with a slower Line-by-Line (LBL) model for a set of climatological atmospheric profiles, are used to calculate channel-specific regression coefficients in the training phase. Given these regression coefficients, the fast radiative transfer model can compute transmittances for any other input profile. The parameterization of the transmittances makes the radiative model computationally much more efficient and in principle should not add significantly to the errors generated by uncertainties in the spectroscopic data used by the LBL model on which the fast model is based (Matricardi et al., 2001). The additional uncertainty due to the use of RTTOV-gb instead of a LBL model has been quantified in De Angelis et al. (2016)."

Matricardi, M., Chevallier, F., Tjemkes, S., An improved general fast radiative transfer model for the assimilation of radiance observations. ECMWF Technical Memorandum 345, 2001.

**4) Page 5 line 26: I am not sure what NWPSAF is.**

The acronym "NWPSAF" stands for "Numerical Weather Prediction Satellite Application Facility". We included the extended name in the revised manuscript.

**5) Section 2.1 and 2.4: In general, it is not clear how the measurements are used in the comparison with the model output. I see in section 2.2 that the that the model produces profiles every 3 hours, but I assume the MWRs produce brightness temperatures every few seconds or minutes depending on the scanning geometry. How were the two matched temporally? Were the MWRs brightness temperatures averaged at all? Or are just instantaneous temperatures?**

The observation the closest in time to the AROME 3 hour-forecast or the analysis has been selected for the comparison with the NWP model. We mentioned that at the end of Section 2.1:

"Temporal matching of MWR observations and NWP model forecasts has been obtained by selecting MWR TB records closest in time to the model forecast time (only one observation without any average over several). The following O-B analysis is performed on the sample of temporal match-up observation-model couples."

**6) Page 6 Section 2.4 lines 10-14: I do understand the cloud screening with the IR measurements for zenith observations. But if the IR measurements only look at the zenith most of the time they will not be representative of off-zenith measurements. Most likely there will be clouds at lower elevation angles unless you have some other ways to ensure that the clear-sky condition holds horizon to horizon. Therefore, the off zenith analysis is doubtful.**

The authors confirm that the clear-sky selection with the IR measurements and the 31GHz-standard deviations only refers to zenith observations. We assume that the zenith observations are indicative of sky conditions although we agree with the referee that this may not be fully representative of off-zenith measurements. Ancillary data providing cloud presence at other elevation angles (such as those provided by whole sky imagers) are not available at all the considered MWR sites. In addition, the aim is to present a method that can be applied to any site where a MWR instrument is operated stand-alone. The uncertainty due to residual off-zenith cloud contamination contributes to larger O-B differences at lower elevation angles at non-opaque channels, possibly adding to both bias and standard deviation. The clear-sky selection is strictly valid for zenith observation only; that's why only zenith values are reported in Table 2. However, note that cloud contamination does not affect significantly V-band opaque channels, which are those used at lower elevation for boundary layer temperature profile retrievals.

The following sentence has been included in the revised manuscript:

"Note that both the clear-sky selections with the IR measurements and the 31GHz-standard deviations only refer to zenith observations. Thus, this may not be fully representative of off-zenith measurements. Ancillary data providing cloud presence at other elevation angles (such as those provided by whole sky imagers) are not available at all the considered MWR sites. In addition, the aim of this study is to present a method that can be applied to any site where a MWR instrument is operated stand-alone. The uncertainty due to residual off-zenith cloud contamination may contribute to enhanced O-B differences at lower elevation angles, possibly adding to both bias and standard deviation. However, cloud contamination does not affect significantly V-band opaque channels, which are those used at lower elevation for boundary layer temperature profile retrievals. "

**7) Page 6 Section 3 line 32-35: It is not clear why is the 31.40 GHz channel calibrated with LN2 and not with tip curves. Is this the case for all 6 radiometers?**

HATPRO absolute calibrations at JOYCE were carried out on June 6 and August 8 2014 as LN2 calibrations. Continuous sky tipping calibrations, which also enable absolute calibration of the K-band channels, were disabled in order to provide a homogeneous calibration procedure to all channels and due to the assumed sufficiently high stability of the microwave K-band receivers.
The authors confirm that sky tipping calibration was disabled for all the other radiometers.

**8) Page 7 lines 38-40, Page 8 line 1-18: I don't understand this step at all. Why is this bias now corrected with a bias derived from another model? Why is it necessary to do this? I think the author should explain this step.**

As mentioned in the text (Section 3), the authors are aware that, to be consistent, the bias correction should be computed using the same NWP and radiative transfer model as used for the O-B comparison. This approach is highly recommended for any further use of this dataset. Here, we just want to give a qualitative example of how a bias correction can remove most of the systematic errors at 51-53 GHz. Thus, we used an independent bias correction computed for JOYCE, with respect to another NWP model (COSMO-DE). The comparison shows that even using different NWP and radiative transfer models, a significant improvement in the O-B biases and RMS can be obtained. This seems to suggest that the systematic errors at 51-53 GHz are likely due to calibration issues and/or spectroscopic uncertainties in state-of-the-art microwave absorption models.

We modified the text in Section 3 to make this point clearer.

**9) Page 8 lines 21-34: As already mentioned in the previous comment, are you sure that there are no clouds off zenith? Secondarily scanning the radiometers at low elevation angles (< 20 degrees) has a high chance of contamination of the readings. Even if a radiometer is elevated a few meters above the surface there is the chance of contamination from foreign objects such as electric cables, poles, buildings, mountain tops, antennas, etc. Are the authors sure that the location and installation of the six radiometers is adequate for such low-angle scanning geometry?**

The authors agree with the referee that observations at low elevation angles could suffer from contamination from undetected clouds (see reply to comment 6). The authors are confident that location and installation of the radiometers are appropriate, as they were carefully designed for long term monitoring (e.g. 4 out of 6 sites belong to the reference network GRUAN). In addition, O-B statistics are found to be consistent among the instrumental sites (in particular standard deviations) down to 10° elevation angle; this seems to suggest that no significant site-specific contamination is affecting the comparison.

The following sentence has been included in the revised manuscript:

"Moreover, O-B statistics are found to be consistent among the instrumental sites (in particular standard deviations) down to 10° elevation angle; this seems to suggest that no significant site-specific contamination is affecting the comparison."

**10) Page 9 lines 3-10: This part is not clear. The authors introduce now the difference between observation and analysis. It would be good to explain what are the Analysis data and how they differ from the forecast data. From the subsequent discussion I infer that the Analysis data are the data produces by the model after the assimilation of additional information from measurements (?).**

The following sentence has been added in the revised manuscript:

"The Arome analysis is the result of the blending of the Arome 3-hour forecast with all the observations available at Météo-France (from satellites, radiosondes, surface networks, etc., but not from MWRs) at the same time."

**11) Page 9 Line 8-10: "Thus, forecast and analysis…this may suggest that there is useful information in MWR data from improving NWP data assimilation" I am not sure I follow this reasoning, why the fact**

that the assimilated data didn't produce much difference with the respect to the MWR suggests that the MWR can improve the assimilation?

As mentioned in the text, NWP analyses have been performed by assimilating all the observations available at Météo-France. Assuming the observations as the reference, despite the large number of observations assimilated into the analyses (but not into the 3h-forecasts), O-B and O-A statistics are still similar. This seems to suggest that the assimilated observations did not bring useful information to the analysis with respect to forecast, perhaps because little information is provided where it would be useful (e.g. in the boudary layer). In this perspective, the assimilation of MWR brightness temperatures into NWP may provide useful information in the boundary layer, possibly leading to improvements in forecast skills.

12) Page 10 line 35: "Statistics at K-band increase with decreasing…" Is statistics the right word here?

Agreed. We changed the word "Statistics" with "Bias, standard deviation and RMS" in the revised manuscript.

13) Fig. 5 Caption: Letters A, B, C, etc. are missing in the figure.

Thanks for spotting this typo. Letters have been introduced in Fig. 5 in the revised manuscript.

14) General comment about Figures 3,4,5, and 6: I wonder if instead of having 2 panels (22- 32 GHz, and 50-59 GHz) it would be better to have only one plot for the whole frequency range (22-59 GHz). This would place the differences on the same vertical scale and make it easier to appreciate the differences.

The authors prefer to keep Figures 3, 4, 5, and 6 as they are for the following reasons:

1) Vertical scales of K- and V-Band are the same down to 30° elevation angle.

2) At lower elevation angles (< 19°), the range at K-Band is much larger than the one at V-Band; thus using the same vertical scale would make results at V-band nearly unreadable.

3) No channel is located between 31 GHz and 50 GHz, thus leaving nearly 1/3 of the panel unused in case of single plot for the whole frequency 22-59 GHz range.

[revised manuscript text omitted]
 | 1.588 | 2.032 | 0.647 | | 0.821 | | 0.940 | -0.549 | -2.706 | -2.429 | -1.262 | -0.905 | -0.932 | -0.831 |
| | ±0.122 | ±0.117 | ±0.086 | | ±0.063 | | ±0.060 | ±0.057 | ±0.048 | ±0.031 | ±0.034 | ±0.039 | ±0.039 | ±0.045 |